# Storing quantum coherence in a quantum dot nuclear spin ensemble for over 100 milliseconds

Harry E. Dyte [1], Santanu Manna [2,4], Saimon F. Covre da Silva [2,5], Armando Rastelli [2] & Evgeny A. Chekhovich [3] ✉

States with long coherence are a crucial requirement for qubits and quantum memories. Nuclear spins in epitaxial GaAs/AlGaAs quantum dots are a great candidate, offering excellent isolation from external environments and on-demand coupling to optical flying qubits. However, coherence times are limited to $\lesssim 1$ ms by the dipole-dipole interactions between the nuclei and by the nuclear quadrupolar coupling to inhomogeneous crystal strain. Here, we combine strain engineering of the nuclear spin ensemble and tailored dynamical decoupling sequences to achieve nuclear spin coherence times exceeding 100 ms. Recently, a reversible transfer of quantum information into nuclear spin ensembles has been demonstrated in quantum dots: our results provide a path to develop this concept into a functioning solid-state quantum memory suitable for quantum repeaters in optical quantum communication networks.

Quantum memories are indispensable in large scale quantum networks, which are expected to enable long distance communication of quantum information[1-4]. Quantum memories have several key requirements[2], a primary figure of merit is the storage time, which is directly related to quantum repeater communication distance. Millisecond-range storage time allows for improvements over direct transmission through an optical fibre[5,6]. Another requirement is for the ratio of the storage time and the entanglement generation time, known as quantum link efficiency[7], to be as high as possible. Although entanglement generation time is currently the main limitation[8], its continuous improvement (reduction) highlights the need for even longer storage times, exceeding 100 ms, in order to achieve worldwide optical communication.

The storage of a quantum state in a memory is limited by the coherence time $T_2$. Several material systems offer long $T_2$, ranging from seconds to hours[9], including trapped atomic ensembles[10,11], ions[12-17], electron spins[18,19] and phosphorus nuclear spins[20-22] in silicon, as well as electron and nuclear spins of impurities in diamond[23,24].

However, long $T_2$ are often negated by poor optical properties required for a long-distance quantum network. There are promising hybrid approaches, such as combination of transmon qubits with solid-state quantum memories[25], but these often suffer from coupling inefficiencies and bandwidth mismatch[9] (see further discussion in Supplementary Note 4).

Epitaxial quantum dots (QDs) in group III-V semiconductors have high qubit entanglement rates[26] and are excellent on-demand emitters of single[9,27-29] and entangled photons[30,31]. At the same time, QDs host material qubits: Electron spin qubits can be interfaced with optical photon qubits[32-34], but the coherence of the electron spin is limited to $\approx 100$ μs[35]. The nuclear spins are isolated from external environments, resulting in long lifetimes and coherence times[36-38]. The recent demonstration of a reversible quantum state transfer between an electron spin qubit and a nuclear spin ensemble (with fidelity of $\approx 0.68$)[39,40] offers a route for electron-mediated storage of a photonic quantum state in a nuclear ensemble of a QD. However, since all atoms in group III-V materials have non-zero nuclear spins, the natural

[1]School of Mathematical and Physical Sciences, University of Sheffield, Sheffield, United Kingdom. [2]Institute of Semiconductor and Solid State Physics, Johannes Kepler University Linz, Linz, Austria. [3]Department of Physics and Astronomy, University of Sussex, Brighton, United Kingdom. [4]Present address: Department of Electrical Engineering, Indian Institute of Technology Delhi, New Delhi, India. [5]Present address: Instituto de Física Gleb Wataghin, Universidade Estadual de Campinas (UNICAMP), Campinas, Brazil. ✉ e-mail: E.Chekhovich@sussex.ac.uk

nuclear spin coherence is limited to a rather modest ≈1 ms range[41]. Extending nuclear spin coherence is thus a key task in achieving quantum memories suitable for quantum repeaters[42,43].

Here, we achieve nuclear spin coherence of over $T_2 \approx 100$ ms, made possible by applying two concepts: Firstly, elastic strain is used to engineer the spin-3/2 nuclei and spectrally isolate the subspace with $I_z = \pm 1/2$ spin projections. The small inhomogeneity of this subspace allows application of thousands of coherent control operations, thus enabling efficient dynamical decoupling. Secondly, a dedicated 40-pulse decoupling sequence cycle is designed to extend nuclear spin ensemble coherence while overcoming the parasitic spin locking effects encountered in previous decoupling experiments[44,45]. The analysis of the results shows that residual decoherence is dominated by the finite-pulse effects and the effective three-body nuclear spin interactions, which are often overlooked. We predict that even longer $T_2$ values, on the order of ≈1 s, are well within reach through larger strains and further advances in dynamical decoupling. The macroscopically long coherence times achieved here were previously possible only in group IV semiconductor spin qubits[21,46], where optical efficiency is limited. Our demonstration of engineered long coherence unlocks the unrivalled optical properties of group III-V materials for applications in quantum memory devices.

## Results

### Strain-engineered nuclear spin ensemble

We study the nuclear spin coherence of GaAs/AlGaAs QDs grown by molecular beam epitaxy. The right inset of Fig. 1a sketches the QD nuclear spin system of $N \approx 5 \times 10^4$ nuclei. The three isotopes $^{75}$As, $^{69}$Ga, and $^{71}$Ga all have nuclear spin $I = 3/2$. The sample is cooled to ≈4.2 K. A superconducting magnet is used to apply a static magnetic field $B_z \approx 5.16$ T along the sample growth crystal direction [001], lifting the degeneracy of the four nuclear spin states with spin projections $I_z = \pm 1/2, \pm 3/2$ (left inset in Fig. 1a). Optical pumping with circularly polarized light (Faraday geometry) is used to polarize the

nuclear spins along the static magnetic field[47]. The nuclear spin life-time is typically $T_1 > 10$ s[48], significantly longer than the coherence times measured in this work. Nuclear spin polarization is measured via photoluminescence (PL) spectroscopy[49], see examples in Fig. 2a. A copper coil generates oscillating magnetic field perpendicular to $B_z$, enabling optically detected nuclear magnetic resonance (ODNMR) measurements. Radio frequency (Rf) bursts with raised cosine envelope are used to transfer coherence in and out of the storage nuclear spin subspace $I_z = \pm 1/2$ and to perform its dynamical decoupling.

The sample is stressed uniaxially along the [110] crystal direction, perpendicular to the static magnetic field. The resulting anharmonicity[37,50], is characterised by the first-order nuclear quadrupolar splitting $\nu_Q^{(1)}$. The measured NMR spectrum, shown in Fig. 1a for $^{75}$As nuclei in a neutral (0e) GaAs/AlGaAs QD, reveals $\nu_Q^{(1)} \approx 255.1$ kHz. This splitting significantly exceeds the linewidths of the NMR transitions: the full width at half maximum (FWHM) $\Delta\nu_{+1/2\leftrightarrow+3/2} \approx \Delta\nu_{-3/2\leftrightarrow-1/2} \approx 13.8$ kHz of the satellite transitions (STs) $-3/2 \leftrightarrow -1/2$ and $+1/2 \leftrightarrow +3/2$ is dominated by the inhomogeneous quadrupolar broadening, while the FWHM $\Delta\nu_{-1/2\leftrightarrow+1/2} \approx 0.8$ kHz of the central transition (CT) $-1/2 \leftrightarrow +1/2$ is controlled by a combination of the second-order quadrupolar inhomogeneity and the dipole-dipole interactions[45,51]. The small linewidth, combined with strain-induced spectral isolation from STs, makes the CT an ideal spin subspace for coherence storage. Notably, the lattice-matched GaAs/AlGaAs QDs offer a significant advantage over Stranski-Krastanov QDs characterised by the much larger $\Delta\nu_{-1/2\leftrightarrow+1/2} \approx 10 - 40$ kHz[45].

### Hamiltonian engineering of a nuclear spin ensemble

Dynamical control of spin interactions is a powerful technique in magnetic resonance[52,53]. The method is based on applying a sequence of Rf pulses that perform fast coherent rotations of the spins, separated by the free evolution intervals. In the interaction picture (the "toggling" frame of reference) the Rf pulses can be viewed as

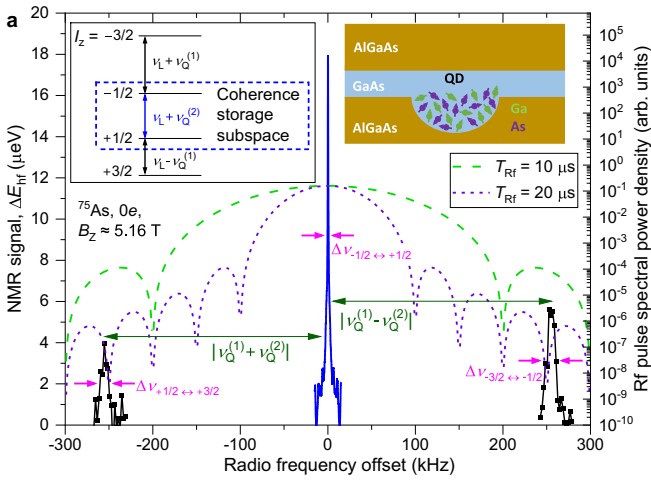

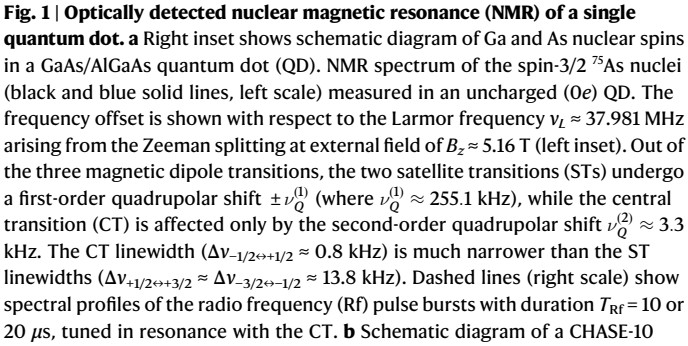

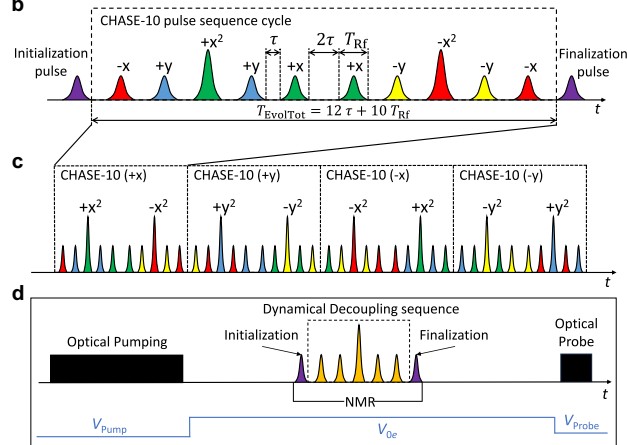

**Fig. 1 | Optically detected nuclear magnetic resonance (NMR) of a single quantum dot. a** Right inset shows schematic diagram of Ga and As nuclear spins in a GaAs/AlGaAs quantum dot (QD). NMR spectrum of the spin-3/2 $^{75}$As nuclei (black and blue solid lines, left scale) measured in an uncharged (0e) QD. The frequency offset is shown with respect to the Larmor frequency $\nu_L \approx 37.981$ MHz arising from the Zeeman splitting at external field of $B_z \approx 5.16$ T (left inset). Out of the three magnetic dipole transitions, the two satellite transitions (STs) undergo a first-order quadrupolar shift $\pm\nu_Q^{(1)}$ (where $\nu_Q^{(1)} \approx 255.1$ kHz), while the central transition (CT) is affected only by the second-order quadrupolar shift $\nu_Q^{(2)} \approx 3.3$ kHz. The CT linewidth ($\Delta\nu_{-1/2\leftrightarrow+1/2} \approx 0.8$ kHz) is much narrower than the ST linewidths ($\Delta\nu_{+1/2\leftrightarrow+3/2} \approx \Delta\nu_{-3/2\leftrightarrow-1/2} \approx 13.8$ kHz). Dashed lines (right scale) show spectral profiles of the radio frequency (Rf) pulse bursts with duration $T_{Rf} = 10$ or 20 μs, tuned in resonance with the CT. **b** Schematic diagram of a CHASE-10

sequence cycle, letters and signs denote Rf pulse phases. The pulses are separated by the free-evolution intervals τ. The total nuclear evolution time is $T_{EvolTot} = 10T_{Rf} + 12\tau$, while $T_{FreeEvol} = 12\tau$ is the pure free evolution time for one cycle. **c** The CHASE-40 supercycle constructed of four CHASE-10 steps, with pulse carrier phase incremented by $\pi/2$ in each step. **d** Timing of the ODNMR measurement cycle. Optical pumping creates longitudinal nuclear spin polarization. The initialization $\pi/2$ Rf pulse converts this into transverse (coherent) nuclear polarization in the $xy$ plane. Dynamical decoupling is applied, followed by a finalization $\pi/2$ pulse to rotate the remaining transverse polarization back along the z-axis. Finally, the nuclear polarization is read out using photoluminescence (PL) spectroscopy under an optical probe pulse. The sample bias is pulsed to maximize optical nuclear spin pumping and PL intensity during optical probing.

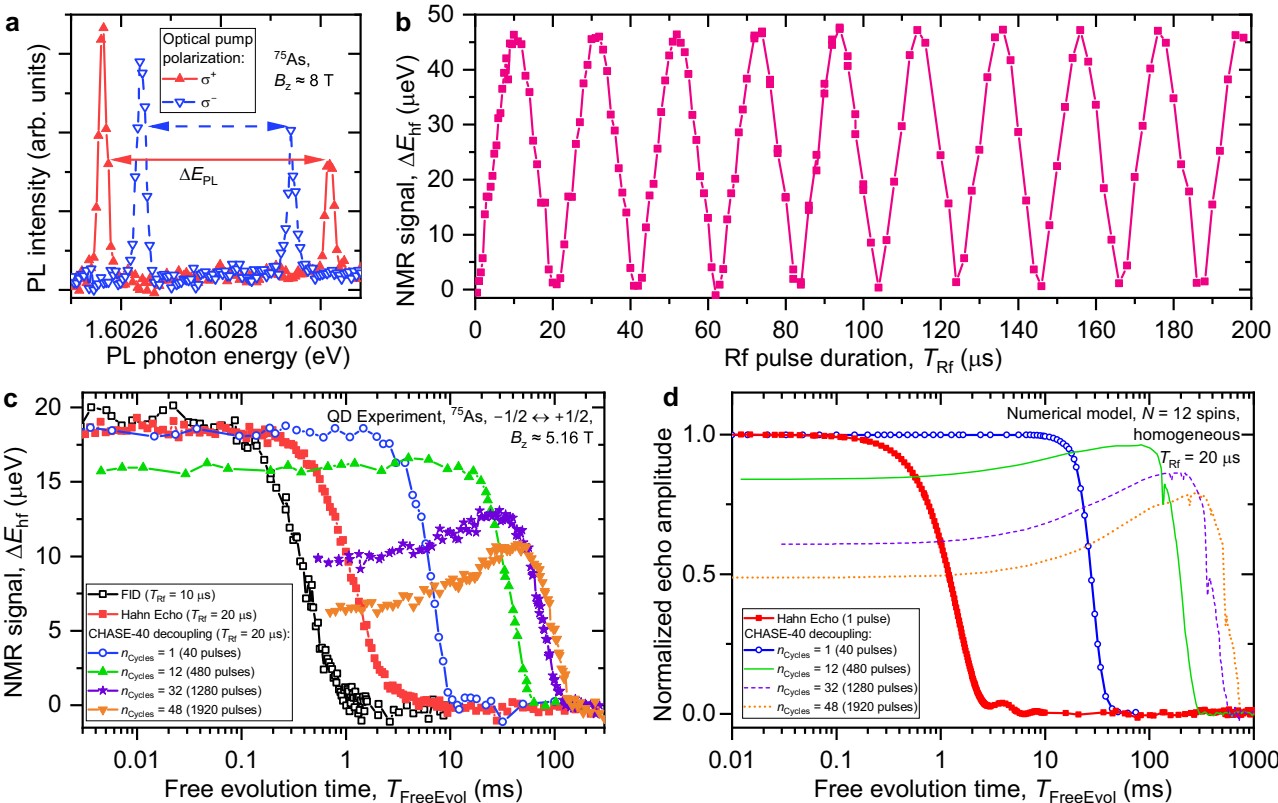

**Fig. 2 | Dynamical decoupling of QD nuclear spins. a** Two photoluminescence (PL) spectra of a neutral exciton in the same individual QD measured after optical pumping with $\sigma^+$ ($\sigma^-$) polarized light, which results in negative (positive) nuclear spin polarization. The PL spectral splitting $\Delta E_{PL}$ is a sum of the constant Zeeman splitting and the nuclear hyperfine shift $\Delta E_{hf}$, which is derived from the variations of $\Delta E_{PL}$. **b** Rabi oscillations of the nuclear spins in a neutral (0e) QD observed under an increasing duration $T_{Rf}$ of an Rf pulse of constant amplitude. **c** Nuclear spin decoherence measured by optically detected nuclear magnetic resonance

(ODNMR) under free induction decay (FID, open squares), Hahn echo (solid squares) and an increasing number of CHASE-40 dynamical decoupling cycles $n_{Cycles} = 1 - 48$ (see legend). Rf pulses, with duration $T_{Rf} = 20\,\mu s$, are applied to the central spin transition $-1/2 \leftrightarrow +1/2$ in a neutral (0e) QD. Nuclear spin polarization is initialized with an x pulse ($\phi = 0$). **d** Numerical modelling of nuclear spin decoherence on a homogeneous ensemble of $N = 12$ spins under the same decoupling sequences as in (**c**).

transformations of the spin-interaction Hamiltonian. The $\pi$-pulse rotations invert the sign of the frequency shifts, allowing refocusing of the dephasing[54], which in QDs is caused primarily by inhomogeneous quadrupolar broadening. However, sequences of $\pi$ pulses, such as Carr-Purcell[55] or XY8[56], do not recouple the nuclear spin-spin dipolar interactions. Instead, these dipolar interactions can be averaged to zero with a sequence of four phase-shifted $\pi/2$-pulses[57]. The average Hamiltonian is the leading (0th order) term in the Magnus expansion of the entire Hamiltonian in the toggling frame. By introducing more complex sequences of pulses it is possible to eliminate the unwanted interactions to higher orders, thus engineering the spin Hamiltonian to have the desired form[53].

Here, we engineer a "time suspension"[58] type of sequence, where the Hamiltonian terms are eliminated as much as possible to preserve an arbitrary coherent state of the nuclear spin ensemble for the longest possible time. As a starting point we use a CHASE-10 cyclic sequence[45] shown in Fig. 1b. By combining $\pi/2$ and $\pi$ pulses, this sequence eliminates the average (0th order) free-evolution Hamiltonian both for the resonance frequency shifts and the spin-spin interactions. Simultaneous suppression of both types of interactions is crucial for dynamical decoupling in a dense nuclear spin ensemble of GaAs. By symmetrizing the sequence, a CHASE-20 supercycle is formed, which further eliminates all the 1st-order terms in the Hamiltonian. The CHASE-20 sequence has been applied to QDs previously, demonstrating its ability to suppress decoherence even under large inhomogeneous resonance broadenings in Stranski-Krastanov InGaAs/GaAs QDs[45]. In low-strain GaAs/AlGaAs QDs, nuclear spin coherence

times up to $T_2 \approx 20$ ms have been achieved[37]. However, the $\pi$ pulses cause spin locking[44,45] which selectively accelerate decoherence of the spin states polarized along a certain equatorial axis of the Bloch sphere in the rotating frame, while artificially enhancing ("locking") the coherence of the states polarized along the orthogonal equatorial axis. This behaviour is unwanted in quantum memory applications as it may lead to distortion of the state during storage.

Here we use a different approach, where four CHASE-10 cycles are combined into a CHASE-40 supercycle shown in Fig. 1c. The phases of the Rf pulses in each CHASE-10 subcycle are stepped by $\pi/2$. While each subcycle causes spin locking, the preferential direction of the "lock", when viewed in the rotating frame, makes a full rotation about the direction of the static magnetic field ($z$) over the CHASE-40 supercycle. This four-step "rotating spin lock" eliminates the net spin locking effect for an arbitrary coherent state, as demonstrated through rigorous calculation (See Supplementary Note 5). Furthermore, the leading order residual Hamiltonian of CHASE-40 is twice smaller than in CHASE-20, resulting in extended coherence.

## Extended spin coherence under dynamical decoupling
We start by examining experimentally the nuclear spin dynamics under continuous resonant Rf driving. The results shown in Fig. 2b reveal Rabi oscillations, which confirm the coherent nature of spin driving and allow the $\pi/2$ and $\pi$ Rf pulses to be calibrated for dynamical decoupling (See Supplementary Note 2C). We then apply dynamical decoupling to the isolated $I_z = \pm 1/2$ nuclear spin subspace with two varying parameters: the number of sequence cycles $n_{Cycles}$ and the

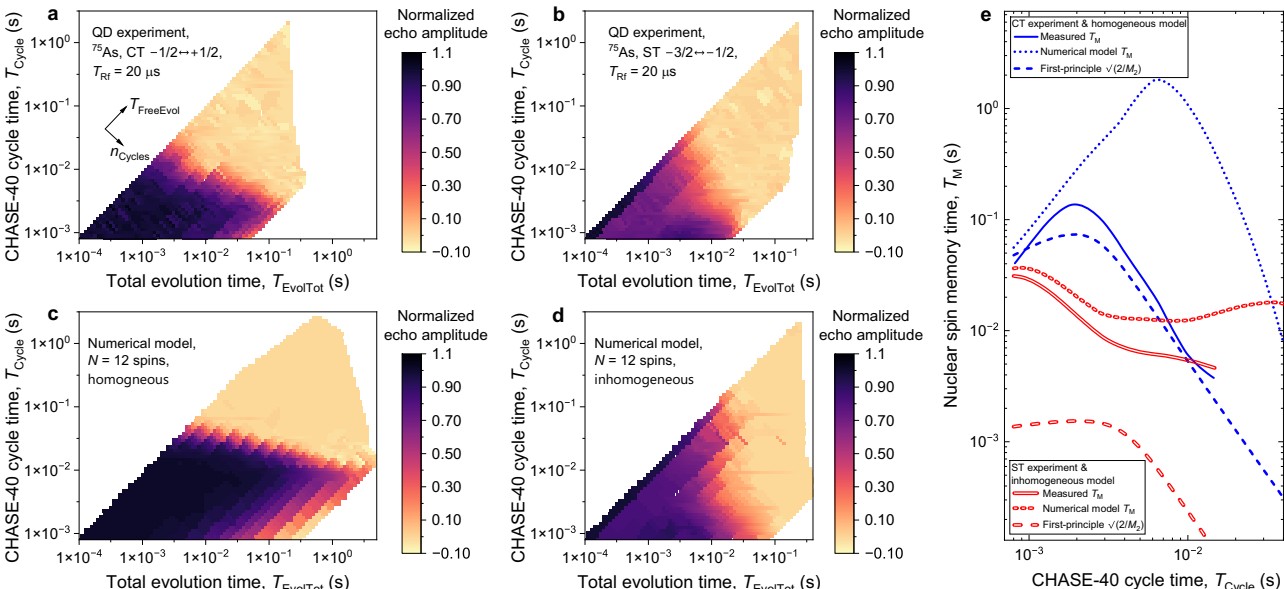

**Fig. 3 | Coherence storage in a nuclear spin ensemble under dynamical decoupling. a** Decoherence of the $I_z = \pm 1/2$ subspace of the $^{75}$As nuclei measured in a neutral ($0e$) QD under dynamical decoupling with Rf pulse duration of $T_{Rf} = 20\,\mu s$. The normalized echo amplitude (colour scale) is shown as a function of the total evolution time $T_{EvolTot} = n_{Cycles}T_{Cycle}$ (free evolution plus Rf pulses, horizontal axis) and the rate of Rf pulsing expressed in terms of the duration $T_{Cycle} = 40T_{Rf} + 48\tau$ of a single CHASE-40 cycle (vertical axis). The axes of the plot correspond to a nonlinear transformation from the variables $n_{Cycles}$ and $T_{FreeEvol} = 40\tau n_{Cycles}$ used in Fig. 2c. The plot combines data obtained with different numbers of CHASE-40 cycles and with shorter subcycles such as CHASE-10. **b** Decoherence of the $I_z = (-3/2, -1/2)$ subspace of the $^{75}$As nuclei measured under dynamical decoupling with $T_{Rf} = 20\,\mu s$.

**c** Decoherence under dynamical decoupling derived from first-principles numerical modelling of spin dynamics of a homogeneous ensemble of $N = 12$ nuclei. **d** Numerically modelled decoherence of $N = 12$ nuclei subject to inhomogeneous spectral broadening. **e** Nuclear spin memory time $T_M$ derived from exponential fitting of decoherence measured as a function of $T_{EvolTot}$. Single lines show results for CT experiments and numerical modelling with the $I_z = \pm 1/2$ homogeneous subspace, while double lines show results for ST experiments and $I_z = (-3/2, -1/2)$ inhomogeneous subspace. Solid lines show fitting of experimental data, dotted lines show fitting of the data from numerical modelling, dashed lines show $T_M$ calculated analytically from the second moment $M_2$ of the residual Hamiltonian under dynamical decoupling.

total free evolution time $T_{FreeEvol}$. Figure 2c shows nuclear spin coherence decay measured using ODNMR. In the simplest case of free induction decay (FID), without any dynamical decoupling (open black squares), the dephasing time is $T_2^* \approx 0.46$ ms and is a combined effect of quadrupolar inhomogeneity and dipole-dipole interactions. By using a single $\pi$ pulse as a decoupling sequence (solid squares), we find the Hahn echo[54] coherence time $T_2^{HE} \approx 1.38$ ms, which is dominated by the dipole-dipole interactions.

The measured effect of dynamical decoupling with one cycle of CHASE-40 is shown by the circles in Fig. 2c. The decay of the transverse nuclear polarization is plotted as a function of the total free evolution time $T_{FreeEvol}$ (i.e. excluding the duration of the Rf pulses), and reveals a significant extension of the coherence time $T_2^{1\times CHASE-40} \approx 12.3$ ms. With the increasing number of CHASE-40 cycles the coherence time is extended further, reaching $T_2^{48\times CHASE-40} \approx 106.6$ ms for $n_{Cycles} = 48$ (orange triangles), an improvement by 2 orders of magnitude compared to the bare Hahn echo coherence time. The achieved nuclear spin coherence is also 3 orders of magnitude longer than the coherence of a dynamically decoupled electron spin in these QDs[35]. These results demonstrate the superior properties of the nuclear spins as a coherence-storage medium.

It can be seen from Fig. 2c that under an increasing number of decoupling cycles $n_{Cycles}$ the coherence (the NMR echo signal) is reduced even in the limit of $T_{FreeEvol} \to 0$. Moreover, the dependence of transverse nuclear spin polarization on $T_{FreeEvol}$ becomes non-monotonic. These are the indications of nuclear spin decoherence during the finite (nonzero duration) Rf pulses. For quantum memory applications, we are interested in minimising decoherence, whether caused by the Rf control pulses or spin interactions during free evolution. We seek this optimum by replotting the data of Fig. 2c in Fig. 3a, where the normalized nuclear spin coherence is shown as a function of

the total evolution (free evolution plus Rf pulses) time $T_{EvolTot}$ (horizontal axis) and the duration $T_{Cycle}$ of one CHASE-40 cycle (vertical axis). The individual decay plots of Fig. 2c measured at fixed $n_{Cycles}$ now appear along the diagonal lines in Fig. 3a. We fit the decay of coherence with an exponential function of $T_{EvolTot}$: the resulting decay time, which we denote as spin memory time $T_M$, is distinct from the coherence time $T_2$ and is shown as a function of $T_{Cycle}$ by the single solid line in Fig. 3e. The maximum $T_M \approx 136$ ms is achieved not under the fastest possible Rf pulsing, but at $T_{Cycle} \approx 2$ ms, which is 5 times longer than the minimum $T_{Cycle} \approx 0.8$ ms achieved at $T_{Rf} = 20\,\mu s$. The non-monotonic dependence $T_M(T_{Cycle})$ also manifests in non-monotonic decay of spin coherence when plotted as a function of $T_{FreeEvol}$ ($n_{Cycles} = 32$ and 48 shown in Fig. 2c). We conclude that the finite-pulse effects are the main limitation to extending coherence storage through fast dynamical decoupling.

Decoherence during the Rf pulses can in principle be suppressed by reducing $T_{Rf}$. However, experiments conducted on the $I_z = \pm 1/2$ spin states with a reduced $T_{Rf} = 10\,\mu s$ yield faster decoherence than under $T_{Rf} = 20\,\mu s$ (see Supplementary Note 3A). This seemingly contradictory result is understood by considering the spectral profiles of the Rf pulses (dashed lines in Fig. 1a). While the unwanted spin decoherence is indeed suppressed under the shorter $T_{Rf} = 10\,\mu s$ pulses, their broader spectral profile results in a stronger overlap with the STs. Such overlap leads to a faster "leakage" of coherence from the storage $I_z = \pm 1/2$ subspace into the $I_z = \pm 3/2$ states. Thus, there is an optimal pulse duration that balances the finite-pulse and the leakage effects. For the studied structure with the strain-induced quadrupolar splitting of $\nu_Q^{(1)} \approx 255.1$ kHz, this optimum is close to $T_{Rf} = 20\,\mu s$. Increasing elastic strain from $\approx 0.0025$ in the studied sample to the $\approx 0.01$ range[59] is a promising route for applying shorter Rf pulses and a further

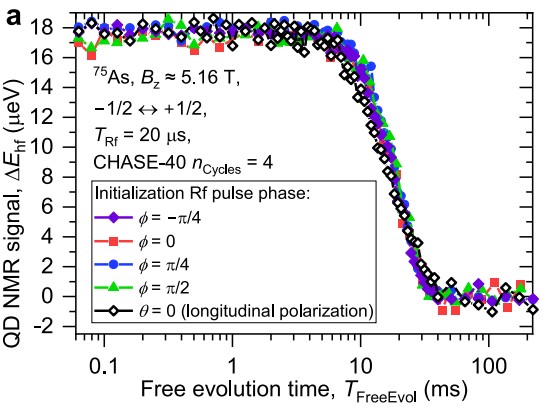
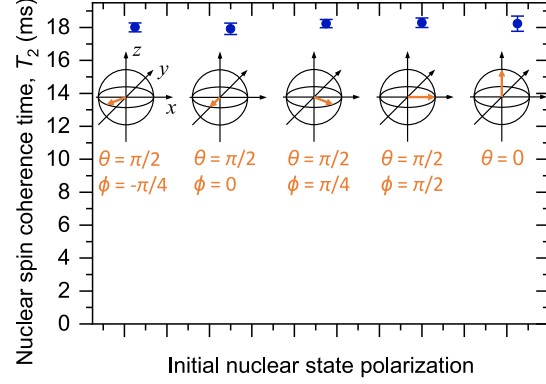

**Fig. 4 | Uniform decoupling of an arbitrary coherent nuclear spin state.**
**a** Nuclear spin decoherence measured under 4 cycles of CHASE-40 with different phases $\phi$ of the initialization Rf pulse, which initializes transverse nuclear polarization along different azimuth angles in the $xy$ plane ($\theta = \pi/2$). The measurement of the longitudinal nuclear spin relaxation under CHASE-40 (without the initialization

pulse, $\theta = 0$) is shown by the open diamonds. **b** The nuclear spin decay times $T_2$ and $T_1$ obtained from fitting the data in (**a**). Error bars are 95% confidence intervals. Schematics show orientation of the nuclei on the Bloch sphere after the initialization pulse, where present.

significant improvement of the storage time and fidelity in a QD nuclear spin quantum memory.

## CHASE decoupling of an inhomogeneous spin ensemble

In order to demonstrate the importance of isolating the homogeneous $I_z = \pm 1/2$ CT storage subspace, we examine the opposite case of a $I_z = (-3/2, -1/2)$ ST subspace. The considerably larger inhomogeneous broadening is characterised by the spectral shape of the ST NMR transitions. For each ST, this is a weighted sum of a peak observed in Fig. 1a with a linewidth of $\Delta\nu_{+1/2\leftrightarrow+3/2} \approx \Delta\nu_{-3/2\leftrightarrow-1/2} \approx 13.8$ kHz (65% weight), and a much broader peak (35% weight), that requires a different NMR technique to be observed. That broad spectral component is not shown in Fig. 1a, but was measured previously[35] to stretch to $\approx \pm 100$ kHz and is caused by the atomic-scale strain of the randomly positioned Al and Ga atoms. The measured ST decoherence under CHASE decoupling is shown in Fig. 3b, and the resulting spin memory time $T_M$ is shown by the double solid line in Fig. 3e. Unlike with $I_z = \pm 1/2$, the best possible dynamical decoupling of the $I_z = (-3/2, -1/2)$ subspace is achieved at the shortest possible $T_{Cycle}$. Despite this fastest possible Rf pulsing, the maximum achieved memory time is $T_M \approx 31$ ms. Although this storage time is a factor of $\approx 15$ improvement over the simple Hahn echo, it is a factor of $\approx 4.4$ worse than $T_M$ achieved for the spectrally narrow $I_z = \pm 1/2$ subspace.

The inferior spin memory time for an inhomogeneously broadened spin ensemble demonstrates how CHASE-40, as any other dynamical decoupling protocol, reaches the limit of its performance when the interaction that is being decoupled is no longer a small perturbation. The inhomogeneous broadening of the $I_z = \pm 1/2$ CT subspace is a small perturbation, characterized by $\Delta\nu_{-1/2\leftrightarrow+1/2}T_{Rf} \approx 0.015 \ll 1$. By contrast, the relative broadening of the $I_z = (-3/2, -1/2)$ subspace is comparable to unity for the observed part of the ST NMR peak ($\Delta\nu_{-3/2\leftrightarrow-1/2}T_{Rf} \approx 0.28$) and violates the perturbative approximation ($\Delta\nu_{-3/2\leftrightarrow-1/2}T_{Rf} > 1$) for the broad component of the ST. The large inhomogeneity of the ST exacerbates decoherence through spin evolution during the finite Rf pulses. Moreover, $\Delta\nu_{-3/2\leftrightarrow-1/2}T_{Rf} \gtrsim 1$ means that Rf control pulses become more "soft" (i.e. not infinitely broad spectrally), resulting in imperfect rotations of the nuclear spins[60,61]. On the other hand, CHASE-40 shows no sign of spin-locking even for the $I_z = (-3/2, -1/2)$ subspace, despite its larger inhomogeneous broadening (See Supplementary Note 3B).

## Uniform decoupling of an arbitrary coherent nuclear spin state

An ideal quantum memory must store any given state with equally high fidelity. However, dynamical decoupling can create parasitic spin

locking regimes, where storage effectiveness depends on the initial state[44,45]. We examine the uniformness of the quantum state storage by measuring dynamical decoupling of the homogeneous $I_z = \pm 1/2$ subspace with different initial states. We use CHASE-40 with $n_{Cycles} = 4$ cycles, which provides a balance between extending the coherence time by an order of magnitude, while keeping to a minimum the finite-pulse decoherence. The phase $\phi$ of the initialization Rf pulse is varied to prepare transverse nuclear spin polarization along the different axes in the equatorial $xy$ plane of the rotating frame ($\phi = 0$ corresponds to a "+x" Rf pulse and prepares polarization along the $-y$ axis, a "+y" pulse with $\phi = \pi/2$ prepares polarization along the $x$ axis). We further perform a measurement, where the initialization pulse is omitted, corresponding to initial nuclear spin polarization along the strong magnetic field ($\theta = 0$). The measured decay curves are shown in Fig. 4a. The coherence times obtained from fitting are shown in Fig. 4b, and are around $T_2 \approx 18$ ms, nearly independent of the initial state.

The uniform dynamical decoupling of different initial states confirms experimentally the design principle of the CHASE-40 supercycle. Stepping of the Rf pulse phases between the four constituent CHASE-10 subcycles can be understood intuitively as a "rotary spin lock": the axis of spin locking is slowly precessing with respect to the rotating frame, with a net effect of removing any preferential spin locking axis over the entire CHASE-40 cycle. Rigorous calculations confirm this result, showing that the symmetry axis of the residual Hamiltonian of CHASE-40 is along the strong static magnetic field ($z$ axis). It is also worth noting that the measurement without any initialization pulse ($\theta = 0$ in Fig. 4b) yields the longitudinal spin relaxation time $T_1$, as opposed to the transverse coherence time $T_2$ measured with an initialization $\pi/2$ Rf pulse ($\theta = \pi/2$). In the absence of dynamical decoupling, the strong static magnetic field imposes a pronounced anisotropy with $T_1 > 10$ s[48] much larger than $T_2 \approx 1$ ms. Under CHASE-40 decoupling $T_1$ decreases and $T_2$ increases, converging to very similar values. This confirms that CHASE-40 is a well-balanced time-suspension sequence that eliminates not only the spin locking anisotropy in the $xy$ transverse plane but also the anisotropy of the strong quantizing magnetic field along the $z$ axis.

The rotary spin lock offers a simple and reliable approach for reusing the dynamical decoupling sequences where spin locking is otherwise present[45,62,63]. The robustness of CHASE-40 against spin locking is key to achieving long spin memory times $T_M \gtrsim 100$ ms through repeated cycling (with up to 2400 pulses).

## Predicting dynamical decoupling performance through analytical and numerical modelling

For any dynamical decoupling sequence the effective spin Hamiltonian can be calculated as a Magnus expansion series. However, finding the spin dynamics from a known Hamiltonian is still a difficult problem. This problem is simplified by noting that the exact spin dynamics is related to the exact NMR spectral lineshape through Fourier transform. The NMR lineshape can be approximated as a Gaussian, and its linewidth can be approximated in terms of the second moment $M_2$, which in turn can be found from the Hamiltonian through direct calculation. The coherent memory time can then be approximated as $T_M \approx \sqrt{2/M_2}$[53]. The residual Hamiltonian of CHASE-40 has been computed analytically up to second order: it is too bulky to reproduce in full, a more detailed discussion can be found in Supplementary Note 5. The Hamiltonian depends on two parameters: the magnitude of the dipole-dipole interaction and the quadrupolar inhomogeneity. These parameters are derived from FID and Hahn Echo experimental data, allowing $T_M^{CHASE-40}$ to be calculated up to second order in analytical form and without any fitting parameters. The results are shown by the dashed lines in Fig. 3e. For the homogeneous subspace $I_z = \pm 1/2$ (single dashed line), the analytical model accurately predicts the peak in the spin memory time $T_M$ at $T_{Cycle} \approx 2$ ms. The actual peak value of $T_M$ is underestimated but matches the experiment within a factor of $\approx 2$. By contrast, for the inhomogeneous subspace $I_z = (-3/2, -1/2)$, the model underestimates $T_M$ by an order of magnitude. This indicates the limited accuracy of the perturbative Magnus expansion, which breaks down when the inhomogeneous broadening (in frequency units) is no longer small in comparison to the reciprocal cycle time $1/T_{Cycle}$. In principle, the analytical model could be improved by extending the Magnus expansion. However, the increasing complexity of the high-order terms makes this approach impractical.

The advantage of the analytical model is that it allows insights into the underlying physics. In particular, we find that direct dipole-dipole interactions of the $i$th and $j$th spins of the ensemble, characterised by the coupling constant $v_{ij}$, is eliminated by CHASE-40 up to second order inclusive. However, the dipolar interaction of the spins $i, j$ remains, but with a coupling constant $\propto v_{ik} v_{kj}$, where $k \neq i, j$ is any other spin. Such a term can be interpreted as an effective three-particle coupling, where the interaction of any two spins $i$ and $j$ is mediated by any other spin $k$. Although often ignored, here we find that the residual decoherence of the homogeneous subspace $I_z = \pm 1/2$ can be explained only by taking into account this effective three-body interaction. The three-body second-order interaction limits $T_M$ under slow dynamical decoupling (long $T_{Cycle}$). In the opposite limit of fast decoupling (short $T_{Cycle}$) the memory time $T_M$ is limited by the zero-order dipole-dipole term arising from spin evolution during the finite Rf pulses ($T_{Rf} > 0$). A combination of these two effects results in a non-monotonic dependence $T_M(T_{Cycle})$ with a maximum in $T_M$, as shown by the single lines in Fig. 3e. By contrast, the decoherence in the inhomogeneous subspace $I_z = (-3/2, -1/2)$ is dominated by the quadrupolar offset inhomogeneity under finite ($T_{Rf} > 0$) control pulses.

We analyse the data from an alternative perspective by conducting numerical modelling of the CHASE dynamical decoupling. The exact Schrödinger equation of a system of $N = 12$ spins is solved using mixed initial spin states with large transverse polarization, mimicking the NMR experiments (see details in Supplementary Note 6). Selected results are shown in Fig. 2d and reproduce well all the main features of experimental results in Fig. 2c. The detailed results for the case of the inhomogeneous subspace $I_z = (-3/2, -1/2)$, are shown in Fig. 3d. Since the decoherence rate is dominated by the quadrupolar offsets, which is a single-particle effect, a good quantitative agreement with the experiment (Fig. 3b) is obtained by using realistic values of the inhomogeneous quadrupolar shifts in the numerical model. The results for the $I_z = \pm 1/2$ subspace, where quadrupolar inhomogeneity is taken to be zero, are shown in Fig. 3c. The numerical model reproduces the

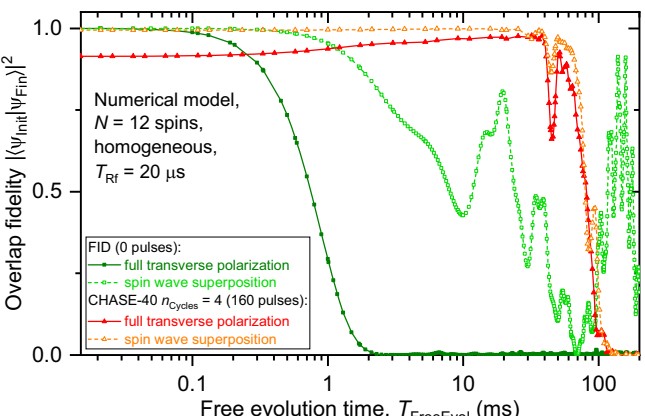

**Fig. 5 | Decoupling of coherent nuclear spin wave states.** Decoherence under dynamical decoupling derived from first-principles numerical modelling of spin dynamics of a homogeneous ensemble of $N = 12$ nuclei. Overlap probability between the initial and final wavefunctions is shown as a function of the free evolution time $T_{FreeEvol}$. Results are shown for free induction decay (FID, squares) and $n_{Cycles} = 4$ cycles of CHASE-40 (triangles). The spin ensemble is initialized into a coherent state either with transverse polarization (multiple-quantum coherence, solid symbols) or a spin wave (single-quantum coherence, open symbols).

main features of the experimental data on the $I_z = \pm 1/2$ subspace (Fig. 3a), in particular the nonmonotonic dependence of the spin memory time $T_M$ on the decoupling sequence cycle time $T_{Cycle}$. However, the agreement is only within an order of magnitude: the numerically-simulated maximum $T_M \approx 2$ s occurs at $T_{Cycle} \approx 8$ ms, compared to the measured maximum $T_M \approx 0.136$ s at $T_{Cycle} \approx 2$ ms (Fig. 3e). This discrepancy may seem unexpected, given that the numerically-simulated coherence time under simple Hahn echo $T_2^{HE} \approx 1.4$ ms is very close to the measured $T_2^{HE} \approx 1.38$ ms. However, the decoherence under Hahn echo is governed by the direct (pairwise) dipole-dipole interaction of the nuclear spins, whereas decoherence under CHASE-40 is dominated by the residual effective three-spin interaction. We therefore ascribe the discrepancy in $T_M$ to the limited number of spins in the numerical model: the number of three-spin combinations contributing to decoherence in an ensemble with $N = 12$ is considerably smaller than in a real crystal lattice of a QD (see Supplementary Note 6). The discrepancy is also likely to include the small but nonzero quadrupolar inhomogeneity of the $I_z = \pm 1/2$ subspace, which reduces CHASE-40 $T_M$ in a real QD.

## Decoupling of coherent spin wave states

The dynamical decoupling experiments of this work are conducted on nuclear spin states with macroscopic transverse polarization, which corresponds to multiple-quantum coherence. By contrast, recent proposals for nuclear-spin-based quantum memories[40] rely on single-quantum spin wave (magnon) nuclear spin states. The applicability of CHASE-40 dynamical decoupling to the spin wave coherent states is a non-trivial question, which we now investigate using numerical modelling on an ensemble of $N = 12$ spins. We compare two types of initial wavefunction states $\psi_{Init}$. One is a state with full transverse polarization (macroscopic magnetization), while the other is a superposition of a ground state with full longitudinal polarization and a spin wave excited state. The spin wave is a single-quantum excitation of the ground state[64] (see details in Supplementary Note 6). The Schrödinger equation is propagated to find the final state $\psi_{Fin}$ of the spin ensemble, and the overlap probability is calculated as $|\langle \psi_{Init} | \psi_{Fin} \rangle|^2$ with results shown in Fig. 5.

The squares in Fig. 5 show the evolution without dynamical decoupling (free induction decay). The state with full transverse polarization exhibits a Gaussian decay on a timescale of $T_2^{HE} \approx 0.9$ ms

(full squares), which is a close match to $T_2^{HE} \approx 1.4$ ms computed above for an initial mixed state with partial transverse polarization. Interestingly, the decoherence of the spin wave superposition (open squares) is considerably slower even without any active dynamical decoupling. Some periodic oscillations and revivals are observed and can be ascribed to the smallness of the spin ensemble ($N = 12$) combined with the pure nature of the initial state. Next we model spin ensemble evolution as a function of $T_{FreeEvol}$ under dynamical decoupling with a fixed number $n_{Cycles} = 4$ of CHASE-40 cycles (triangles in Fig. 5). Once again, the decay exhibits periodic partial revivals, both for macroscopically polarized (full triangles) and spin wave (open triangles) coherent states. Most importantly, compared to free induction decay, CHASE-40 decoupling is found to slow down the decoherence both for macroscopically polarized and single-quantum spin wave coherent states. The computed coherence times $T_2^{4 \times CHASE-40} \approx 90$ ms for pure initial states are found to be very similar to coherence times calculated above for mixed initial states. Notably, the spin wave state is more robust against the finite-pulse decoherence (limit of short $T_{FreeEvol}$) than the macroscopically polarized state. These results suggest that numerical modelling and NMR experiments on spin states with macroscopic transverse polarization can be used to predict the dynamics of a single-quantum spin wave excitation. This initial finding confirms the prospect of using dynamical decoupling in spin wave quantum memories, which will be explored further in future work.

## Discussion

We have demonstrated very long coherence storage times of over 100 ms, achieved despite the dense nature of the nuclear spin ensemble in GaAs, where coherence extension via isotope enrichment[21,46] is not possible. The optically-active GaAs QDs are a promising candidate for quantum memory, which integrates a spin qubit with a single photon sources[9], thus avoiding the need for complex hybrid schemes[25]. The long-term preservation of nuclear spin coherence demonstrated here is a key step in bringing the concept of QD-based optical quantum memory[40] to practical implementation. The extended coherence is enabled by strain engineering of the nuclear spin ensemble and the tailored 40-pulse time-suspension decoupling sequence.

We use a three-pronged approach to the design of dynamical decoupling sequences. Numerical modelling can predict the overall performance of a dynamical decoupling protocol, but its accuracy is limited by the small number of spins $N$, constrained in turn by the exponential scaling of the required classical computing resources with increasing $N$. As a result, numerical simulations are time-consuming: full datasets of Fig. 3c, d require many days of computations on a workstation PC, which is comparable to experimental time required for Fig. 3a, b. Analytical calculations provide good predictions in case of small inhomogeneity. However, for a sequence with 40 pulses, derivation of the residual Hamiltonian takes several hours of computer-assisted algebraic derivations and further tedious manual work to analyse the bulky analytical results. Thus, the two modelling approaches encounter their different limitations, leaving experiment as the ultimate verification of the excellent coherence protection achieved with CHASE-40. The sequence is robust against spin locking, imperfections in control pulses, and is applicable to spin ensembles with a substantial inhomogeneous broadening.

Strain engineering is a key enabling technique, as it allows spectral isolation of the homogeneous $I_z = \pm 1/2$ nuclear spin subspace. A further increase of elastic strain by a factor of $\approx 4$ is within the yield strain of GaAs and is feasible using membrane microstructures[59,65,66]. This would allow quadrupolar splitting in excess of $\nu_Q^{(1)} \gtrsim 1$ MHz, enabling further improvement in quantum memory storage time and fidelity. More importantly, the MHz-range quadrupolar splitting would be required to exceed the electron-nuclear hyperfine interaction, which is $\lesssim 200$ kHz in the studied GaAs QDs[50]. While dynamical decoupling of nuclear spins in presence of the central electron spin qubit is possible in principle[38], achieving long nuclear spin coherence would require spectral isolation (through increased strain) of the hyperfine-broadened nuclear spin transitions.

The maximum storage time $T_M$ under CHASE-40 decoupling is limited by the finite duration of the control pulses, which causes a drop in $T_M$ in the limit of frequent Rf pulsing (single lines in Fig. 3e, limit of small $T_{Cycle}$). By eliminating the zero-order finite-pulse effects[52,67] it should be possible to achieve $T_M \approx 1$ s even at the current level of elastic strain, limited only by the second-order three-particle spin-spin interactions. More broadly, dynamical decoupling can be used to study spin-spin entanglement, thermalization in disordered quantum systems, and many-body localization[68,69].

## Data availability

The data generated in this study are provided in the Source Data file SourceData.zip. Additional information and data related to this study are available from the corresponding author upon request. Source data are provided with this paper.

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

## Acknowledgements

H.E.D. was supported by an EPSRC doctoral training grant. E.A.C. was supported by a Royal Society University Research Fellowship, the Leverhulme Trust grant RPG-2023-141, and the EPSRC awards EP/V048333/1 and EP/V048333/2. A.R. acknowledges support of the

Austrian Science Fund (FWF) via the Research Group FG5, the cluster of excellence Quantum Science Austria (10.55776/COE1), the Linz Institute of Technology (LIT), and the LIT Secure and Correct Systems Lab, supported by the State of Upper Austria, the European Union's Horizon 2020 research and innovation program under Grant Agreements No. 899814 (Qurope), No. 871130 (Ascent+), the QuantERA II project QD-E-QKD and the FFG (grant No. 891366). A.R. and E.A.C. were supported by the QuantERA award MEEDGARD (EPSRC award EP/Z000556/1 and FFG Grant No. 906046). E.A.C. is grateful to Daniel Treuherz, Dorian Gangloff, and Christian Schimpf for fruitful discussions.

## Author contributions

S.M., S.F.C.S. and A.R. developed, grew and processed the quantum dot samples. H.E.D. conducted the experiments. H.E.D. and E.A.C. analysed the data. H.E.D. and E.A.C. drafted the manuscript with input from all authors. E.A.C. performed numerical modelling and coordinated the project.

## Competing interests

The authors declare no competing interests.
