## [Transparent Peer Review file · Nature Communications]

Storing quantum coherence in a quantum dot nuclear spin ensemble for over 100 milliseconds

Corresponding Author: Dr Evgeny Chekhovich

Version 0:

Reviewer comments:

Reviewer #1

(Remarks to the Author)

This paper reports on comprehensive experimental study of coherence of the nuclear spin system in GaAs/AlGaAs quantum dots. The authors combine strain engineering approach to select the 1/2 subspace of the nuclear spins and the tailored protocol for dynamical decoupling. This allows them to get remarkably long nuclear spin coherence time of 100 ms. The developed and used CHASE-40 protocol suppresses leaking effects presented in other approaches and extends the coherence time by a factor of five comparing to the Hahn echo protocol. The authors give analytic and numerical modelling of their results and provide in-depth discussion on this bases. Experiments are made with great care to minor details, being based on long term systematic research of the authors on this problem.

Paper is written very clearly, all experimental results are well presented and discussed, and conclusions are convincing. I readily recommend this paper for publication in Nature Communications.

Recommendations:

To give material of the studied GaAs/AlGaAs QDs in abstract as the used approach of strain engineering are the reached times are specific to that.

Reviewer #2

(Remarks to the Author)

In this work Dyte et al. measure nuclear spin coherence times in a lithographically-defined GaAs quantum dot. By utilizing pulse-based NMR (specifically CHASE pulse sequences) they demonstrate that these coherence times can be extended to ~100ms. The authors claim that these results provide a path towards long-lived quantum memories for quantum networking. The paper is well written and technically sound with thoroughly analyzed experimental results. However, prior to recommending publication, I would ask for some more detail regarding the novelty and impact of this work.

In particular, I am concerned that these results constitute a minor advancement compared to [Chekhovich, E. A. et al. "Nuclear spin quantum register in an optically active semiconductor quantum dot", Nature Nanotechnology, 2020]. This prior work utilized a similar quantum dot sample and operated in a very similar experimental regime. The main differences seem to be a minor change in the pulse sequence (a cyclic permutation of pulse phases mitigating issues associated with spin locking) and a 5x improvement in coherence.

Furthermore, I do not believe that the pulse sequences and coherence times characterized here are useful for quantum networking applications. Specifically, the authors measure coherence times using un-entangled nuclear spin states which are globally polarized along the X (or Y) directions. However, for quantum memory applications, the coherence times of nuclear magnon states are relevant (such as those explored in [Hayhurst, M. et al. "A many-body quantum register for a spin qubit", Nature Physics, 2025]). This is a crucial distinction since, I believe, that the CHASE sequences explored here only cancels dipolar interactions for globally polarized states and not magnonic states.

I also have several more minor/technical queries:

- On Page 1 the authors claim that “Recently, two-way quantum state transfer between photon and nuclear spins has been demonstrated”, however, I do not believe that the two cited papers demonstrate transfer of photonic qubits onto nuclear spin memories.
- It would help in the introduction to introduce the “quantum link efficiency” since it is an essential metric for quantum repeaters; it would also help to explain/characterize the trade-off between memory times and entanglement rates.
- On page 2 the acronym ‘CT’ needs to be defined.
- Regarding the experimental results in Figure 2c: could the authors please provide a qualitative explanation for why the coherence initially increases with free evolution time for 32 and 48 cycle cases?
- On page 4 the authors refer to a “broader invisible peak”, could they please be a bit more specific about what they mean by ‘invisible’ here?
- In Figure 3e it would help to additionally plot the numerical simulation results for ease of comparison.
- In Figure 4 the authors chose to utilize a pulse sequence with only 4 cycles and 18ms coherence time. Could the authors provide equivalent plots for their 100ms coherence time sequence.
- On page 6 the authors state that “It is also worth noting that the T_2 decay time of the transverse polarization (measured with an initialization $\pi/2$ pulse) is nearly identical to that of the longitudinal T_1 decay time.” I am a bit surprised by this statement since the T_1 time is $>10s$, whereas the T_2 time is 100ms, could the authors please clarify.
- Also on page 6 the authors state that “We therefore ascribe the discrepancy in T_M to the limited number of spins in the numerical model”. It would be helpful to explore the convergence of the simulated coherence time with increasing numbers of nuclear spins. Could the authors plot the simulated coherence times for $2 \leq N \leq 12$? Have the authors considered using the cluster-correlation expansion (CCE) method to speed up simulations?

Reviewer #3

(Remarks to the Author)

This manuscript proposes and demonstrates a novel approach to constructing a long-lived quantum memory by utilizing a subspace of the nuclear spin ensemble in a quantum dot. Notably, the memory is encoded into a homogeneous subspace by exploiting strain-induced perturbations in the nuclear spins. As a result, intrinsically long coherence times are successfully achieved. Furthermore, a practical dynamical decoupling (DD) protocol optimized for this system is designed, leading to an experimental demonstration of the longest coherence time to date. These results are accompanied by highly precise and extensive theoretical analysis, which adds depth to the understanding of the mechanisms and ensures reproducibility. Overall, this is a high-quality study on both the experimental and theoretical sides and is evaluated as an important advancement toward the practical realization of quantum repeaters. The novel design concept, the supporting data, and the consistency of the theoretical framework all contribute to the impact of this paper.

Minor comments:

1. The fidelity of the memory initialization remains unclear. When storing or retrieving a photonic quantum state in the quantum memory, the initialization rate and DD control fidelity are important aspects. I would appreciate clarification, as much as possible, regarding these points.
2. The motivation for using CHASE is not clearly described. DD sequences such as XY8 are widely used in general, so I would like the authors to provide some explanation of the advantages of CHASE. Since the nuclear spin ensemble system may have a structural compatibility with CHASE, I believe the authors’ perspective on this would deepen the readers’ understanding.

Version 1:

Reviewer comments:

Reviewer #1

(Remarks to the Author)

I am fine with made revisions and recommend this paper for publication.

Reviewer #3

(Remarks to the Author)

The authors have fully and clearly addressed the two minor comments raised in my previous review. The revised manuscript has improved substantially in clarity and completeness, and I am satisfied with the revisions. I therefore recommend the manuscript for publication in Nature Communications.

RESPONSE TO REVIEWERS' COMMENTS

We would like to start by thanking all three Reviewers for the careful reading and the insightful comments that prompted us to conduct some additional studies and improve the manuscript. Below we provide a point-by-point response to all critical comments in their original order. Reviewers' comments are quoted in *italics font*. The changes in the revised manuscript are highlighted in red font.

Reviewer #1:

This paper reports on comprehensive experimental study of coherence of the nuclear spin system in GaAs/AlGaAs quantum dots. The authors combine strain engineering approach to select the 1/2 subspace of the nuclear spins and the tailored protocol for dynamical decoupling. This allows them to get remarkably long nuclear spin coherence time of 100 ms. The developed and used CHASE-40 protocol suppresses leaking effects presented in other approaches and extends the coherence time by a factor of five comparing to the Hahn echo protocol. The authors give analytic and numerical modelling of their results and provide in-depth discussion on this bases. Experiments are made with great care to minor details, being based on long term systematic research of the authors on this problem.

Paper is written very clearly, all experimental results are well presented and discussed, and conclusions are convincing. I readily recommend this paper for publication in Nature Communications.

Recommendations:

To give material of the studied GaAs/AlGaAs QDs in abstract as the used approach of strain engineering are the reached times are specific to that.

We thank the Reviewer for the positive comments. As per Reviewer's suggestion, we have modified the abstract to clarify that the work is done on GaAs/AlGaAs quantum dots.

Reviewer #2:

In this work Dyte et al. measure nuclear spin coherence times in a lithographically-defined GaAs quantum dot. By utilizing pulse-based NMR (specifically CHASE pulse sequences) they demonstrate that these coherence times can be extended to ~100ms. The authors claim that these results provide a path towards long-lived quantum memories for quantum networking. The paper is well written and technically sound with thoroughly analyzed experimental results. However, prior to recommending publication, I would ask for some more detail regarding the novelty and impact of this work.

In particular, I am concerned that these results constitute a minor advancement compared to [Chekhovich, E. A. et al. “Nuclear spin quantum register in an optically active semiconductor quantum dot”, Nature Nanotechnology, 2020]. This prior work utilized a similar quantum dot sample and operated in a very similar experimental regime. The main differences seem to be a minor change in the pulse sequence (a cyclic permutation of pulse phases mitigating issues associated with spin locking) and a 5x improvement in coherence.

The aim of the work published previously [Nature Nanotechnol. 15, 999 (2020)] was to establish epitaxial (optically active) GaAs/AlGaAs quantum dots as a system of choice for spin qubit studies in QDs, as well as to identify the range of possible applications and future directions to explore. Indeed, one of the demonstrations was that nuclear spin coherence can be extended to tens of milliseconds. However, this was discussed only briefly (one paragraph) and primarily in the context of coherent control fidelity (i.e. the number of coherent control gates that can be applied before the onset of decoherence). Possible applications for quantum memories were not discussed there at all.

The distinct advancements of the present work can be summarized as follows:

- Indeed, the maximum achieved coherence times have been extended from ~ 20 ms to ~ 100 ms. To our knowledge, 100 ms is the longest quantum coherence (of any kind, not just nuclear spin) achieved in a GaAs semiconductor device. While even longer spin coherence is possible in other materials, GaAs quantum dots are unique in their quantum optical properties. We believe that achieving record-long coherence in this material system is by itself an important result.
- Equally important is our demonstration that the ~ 100 ms mark is the actual upper limit to nuclear ensemble coherence, at least with the present level of elastic strain ($\sim 0.3\%$). We identify the mechanisms that are responsible for that upper limit (finite-pulse effects, cross-coupling to satellite NMR transitions) and analyse the pathways for increasing coherence further. These conclusions are supported by rigorous first-principles theory (average Hamiltonian theory) and numerical modelling. None of these results have been reported previously.
- Notably, the coherence is extended by nearly two orders of magnitude (from Hahn echo $T_2 \sim 1.38$ ms to a maximum $T_M \sim 136$ ms under dynamical decoupling). In other words, our dynamical decoupling protocol “switches off” $\sim 99\%$ of interactions in a dense interacting 3D spin lattice. We then reveal the physical mechanisms responsible for the remaining $\sim 1\%$, in particular identifying the effective three-particle interactions as the residual coupling that is not suppressed by dynamical decoupling. We believe it is a substantial advancement, since

understanding how to “switch off” interactions in a many-body system is a stepping stone to engineering an arbitrary Hamiltonian, with potential applications in quantum technologies and magnetic resonance spectroscopy.

- We demonstrate here that a four-step “rotating spin lock” technique gives a nearly perfect isotropic dynamical decoupling of an arbitrary coherent state: it not only eliminates the preferential spin locking direction in the transverse direction, but actually eliminates the distinction between longitudinal relaxation (T_1) and decoherence (T_2). Importantly, this concept is proven experimentally (it is not uncommon for dynamical decoupling sequences to show good promise on paper but offer minimal or no advantage in a real-world experimental test). This demonstration has wider applications in dynamical decoupling, beyond quantum dots.
- We present new results showing that dynamical decoupling is effective not only for coherent states with macroscopic polarization, but also for single-quantum wave states, which are of particular relevance in prospective quantum memory applications. This is discussed further in our response to the next point raised by Reviewer.

The scope of the present study is also different. While the 2020 paper focused on identifying GaAs nuclear spins as a promising quantum system, our present work addresses broader questions. These new results will be relevant to dynamical decoupling techniques in systems beyond quantum dots as well as wider applications in magnetic resonance spectroscopy and many-body quantum physics.

We hope this clarifies why we believe our present work to be a significant advancement compared to prior publications.

Furthermore, I do not believe that the pulse sequences and coherence times characterized here are useful for quantum networking applications. Specifically, the authors measure coherence times using un-entangled nuclear spin states which are globally polarized along the X (or Y) directions. However, for quantum memory applications, the coherence times of nuclear magnon states are relevant (such as those explored in [Hayhurst, M. et al. “A many-body quantum register for a spin qubit”, Nature Physics, 2025]). This is a crucial distinction since, I believe, that the CHASE sequences explored here only cancels dipolar interactions for globally polarized states and not magnonic states.

We thank the Reviewer for this important observation. Stimulated by this question, we have conducted numerical modelling of dynamical decoupling for spin-wave (magnon) coherent states. The results are shown in a new Fig. 5 of the revised manuscript. The key findings are

that: CHASE dynamical decoupling is applicable to spin wave states in a sense that it increases the coherence time compared to the no-decoupling case; CHASE decoupling works similarly well for states with multiple-quantum coherence (globally polarized along transverse direction) and for states with single-quantum coherence (spin waves) in a sense that their coherence times under CHASE decoupling are comparable. Based on these observations we conclude that dynamical decoupling of states with macroscopic transverse polarization (performed via numerical modelling or NMR experiments) is a good predictor for dynamical decoupling of single-quantum spin wave excitation. This observation derived from numerical modelling can also be understood from the first principles: dynamical decoupling works through averaging out of the selected interaction terms by switching the Hamiltonian (in a toggling frame of reference) over a period of the pulse sequence cycle. This averaging is a rigorous mathematical procedure (Magnus series expansion summarised in Supplementary Note 5B): it is a genuine alteration of the effective Hamiltonian that does not depend on the nature of the spin state (whether it's a spin wave or macroscopic polarization). In other words, if a certain interaction term is cancelled through dynamical decoupling, it has no effect on spin evolution, regardless of the nature of the spin state. To sum up, our findings confirm the prospect of using dynamical decoupling in spin wave quantum memories and hopefully answer the Reviewer's immediate point.

Our numerical studies on dynamical decoupling of spin waves (magnons) are rather preliminary at this stage. Several interesting observations can be made even from these preliminary results (Fig. 5). For example, we find that the coherence times of the spin wave state is longer than coherence of a macroscopically polarized state, even without any decoupling. Moreover, when CHASE dynamical decoupling is applied, the spin wave state is found to be more robust against the finite-pulse decoherence. This suggests some additional cancelations in the spin interactions, specific to spin waves. There are many outstanding questions that need to be answered (through further numerical studies), and the ultimate experimental verification of dynamical decoupling of spin waves needs to be conducted. This is an ongoing work. We plan to report on these studies in future separate publications to avoid overloading this present manuscript.

I also have several more minor/technical queries:

- *On Page 1 the authors claim that "Recently, two-way quantum state transfer between photon and nuclear spins has been demonstrated", however, I do not believe that the two cited papers demonstrate transfer of photonic qubits onto nuclear spin memories.*

We have corrected the sentence to state that references [38, 39] report on two-way state transfer between an electron spin qubit and the nuclear spin ensemble of a QD. There is a prospect for a two-way transfer of a photonic state, which would require using the electron spin qubit as a mediator.

- *It would help in the introduction to introduce the “quantum link efficiency” since it is an essential metric for quantum repeaters; it would also help to explain/characterize the trade-off between memory times and entanglement rates.*

In the revised manuscript we briefly introduce the quantum link efficiency and state that increasing the ratio of the memory storage time and the entanglement generation time is one of the key requirements for a quantum memory.

- *On page 2 the acronym ‘CT’ needs to be defined.*

The CT acronym, which stands for central transition, is now defined at the first use in the main text on page 2.

- *Regarding the experimental results in Figure 2c: could the authors please provide a qualitative explanation for why the coherence initially increases with free evolution time for 32 and 48 cycle cases?*

The apparent non-monotonic coherence decay in experiments with 32 and 48 cycles is caused by the decoherence that takes place during the finite-duration Rf control pulses. The finite-pulse effects are best demonstrated by replotting the data as a function of the total evolution time T_{EvolTot} and the decoupling cycle time T_{Cycle} , which we do in Fig. 3. By contrast, Fig. 2c shows the as-measured data where the number of decoupling sequence cycles is kept fixed and only the free evolution time T_{FreeEvol} is varied. Varying T_{FreeEvol} alone (horizontal axis in Fig. 2c) corresponds to varying both T_{EvolTot} and T_{Cycle} , which in essence corresponds to moving along the “diagonal” line in the logarithmic $(T_{\text{EvolTot}}, T_{\text{Cycle}})$ coordinates used in Fig 3. Thus, when T_{FreeEvol} is increased (e.g. along the horizontal axis in Fig. 2c) it increases not only the spin evolution time, but also the T_{Cycle} parameter, which controls the balance between the finite-pulse decoherence and decoherence through free evolution.

Some complexity in interpreting the data is inevitable here and arises from the duality of the metrics used to characterise dynamical decoupling sequences. When considering the spin memory performance, it does not matter whether spin decoherence happens through finite-pulse effects or through free evolution of the interacting spins. In this case, it is more convenient to use the memory time T_M as a metric and plot the data in the $(T_{\text{EvolTot}}, T_{\text{Cycle}})$ coordinates (as in Fig. 3). By contrast, when analysing the fundamental properties of the dynamical decoupling sequence (i.e. the order in which the Hamiltonian is switched in the toggling frame) the interaction of spins during free evolution is most important. In that case it is more convenient to use the coherence time T_2 as a metric and plot the data in the $(T_{\text{FreeEvol}}, n_{\text{Cycles}})$ coordinates (as in Fig. 2c).

In the revised manuscript we have added a brief comment to relate the non-monotonic character of the raw data curves in Fig. 2c to the non-monotonic dependence of the spin memory time $T_M(T_{\text{Cycle}})$, which ultimately stems from the finite-pulse decoherence.

• On page 4 the authors refer to a “broader invisible peak”, could they please be a bit more specific about what they mean by ‘invisible’ here?

This point was indeed poorly worded in the original manuscript. The broader peak (arising from random positioning of gallium and aluminium atoms in the crystal lattice) is not invisible of course, but just requires a different NMR spectroscopy technique to be revealed. This technique, dubbed “integral NMR”, was applied previously to a separate piece of the same semiconductor wafer in [Nature Nanotechnol. 18, 257 (2023)]. Since the role of the broad spectral peaks is not central to the topic of the present work, we don’t repeat these integral NMR experiments here, and simply quote the previous study for the key parameters, i.e. the +/-100 kHz spectral extent and the 35% weight of the broad NMR peak. This point is clarified in the revised experiment.

• In Figure 3e it would help to additionally plot the numerical simulation results for ease of comparison.

As suggested, we have added the spin memory times T_M derived from numerical modelling to Fig. 3e (dotted lines). We have also added selected numerical modelling results in Fig. 2d, to simplify comparison with experiments.

• In Figure 4 the authors chose to utilize a pulse sequence with only 4 cycles and 18ms coherence time. Could the authors provide equivalent plots for their 100ms coherence time sequence.

The 100 ms coherence can be achieved by using $n = 48$ cycles of the decoupling CHASE-40 sequence. As we show in Fig. 2c, for such a large number of cycles the finite duration of the control Rf pulses (which is 20 μ s in our experiments, currently limited by the magnitude of crystal strain) already leads to substantial (more than a half) loss of transverse spin magnetization, even in the limit of short free evolution time. Thus, experiments with a large number of cycles do not reveal much information about the anisotropy of decoherence during free evolution, because the coherence is lost even before any free evolution takes place. The interest here is in spin dynamics during free evolution because decoherence during the pulses can in principle be removed by shortening the control pulses. By contrast, the pulse control sequence itself (i.e. the order in which the Hamiltonian is switched in the toggling frame) is best characterized by the free evolution between the pulse. Consequently, we choose $n = 4$ cycles as a compromise between extending the coherence substantially (from ~ 1.38 ms in Hahn echo to ~ 18 ms with 4 cycles of CHASE-40) and minimising the finite-pulse effects. We clarify this point in the revised manuscript.

We do have a dataset comparing dynamical decoupling of the x- and y-polarized initial states with $n=36$ cycles (but no data for the two diagonal directions of the initial polarization). Since all the pulses in the CHASE-40 sequence are either x- or y-, experiments with x- and y-polarized initial states are in fact sufficient to detect any preferential direction. This data is now shown in Supplementary Fig. 7. We find a small asymmetry in the decoherence induced by the parasitic finite-pulse effects. However, the decoherence during free evolution, characterised by $T_2 \sim 84$ ms, is found to be very similar for both the x- and y-polarized initial states even at $n=36$ cycles of CHASE-40. This gives further weight to our conclusions about uniform (isotropic) decoupling of arbitrary coherent spin states, and hopefully satisfies the Reviewer's request.

• On page 6 the authors state that "It is also worth noting that the T_2 decay time of the transverse polarization (measured with an initialization $\pi/2$ pulse) is nearly identical to that of the longitudinal T_1 decay time." I am a bit surprised by this statement since the T_1 time is >10 s, whereas the T_2 time is 100ms, could the authors please clarify.

This point was also poorly worded in the original manuscript. We clarify this in the revised manuscript as follows. In the absence of any dynamical decoupling, the strong static

magnetic field indeed imposes a pronounced anisotropy with the longitudinal spin relaxation time $T_1 > 10$ s significantly exceeding the transverse coherence time $T_2 \sim 1$ ms (Hahn echo). T_1 is probed in an NMR measurement without any initialization pulse ($\theta=0$ in Fig. 4b), while the transverse coherence time T_2 is measured with an initialization $\pi/2$ Rf pulse ($\theta=\pi/2$ in Fig. 4b). Under CHASE-40 decoupling T_1 decreases and T_2 increases, converging to very similar values. This is a sign of an optimally balanced time-suspension sequence that eliminates not only the spin locking anisotropy in the xy transverse plane but also the anisotropy of the strong quantising magnetic field along the z axis.

• Also on page 6 the authors state that “We therefore ascribe the discrepancy in T_M to the limited number of spins in the numerical model”. It would be helpful to explore the convergence of the simulated coherence time with increasing numbers of nuclear spins. Could the authors plot the simulated coherence times for $2 \leq N \leq 12$?

Following the Reviewer’s comment, we have investigated the dependence of the spin ensemble decoherence on the number of spins up to $N=14$. The results are shown in Supplementary Fig. 8 (Supplementary Note 6) and broadly corroborate our initial conclusions. We compare dynamical decoupling under simple Hahn echo and under 4 cycles of CHASE-40 (160 decoupling pulses). In case of Hahn echo, the coherence time follows a clear trend of reduction with an increasing number of spins N . This is expected, since an increasing N means that each spin interacts with a larger number of neighbours, accelerating the decoherence. In case of CHASE-40 decoupling, the computed coherence time shows no clear dependence on N . Since CHASE-40 eliminates the pairwise spin-spin interactions, the resulting spin dynamics are governed by a delicate balance of weak residual three-particle (and possibly higher order) correlations. Our interpretation is that the available number of spins (up to $N=12..14$ in exact modelling) is too small to converge to spin dynamics that are measured in a nuclear spin bath with a much larger number of spins ($N \sim 10000$ in a quantum dot).

Have the authors considered using the cluster-correlation expansion (CCE) method to speed up simulations?

The technical answer to this question is the balance of cost and benefit. Implementing CCE would come at a cost due to the complexity of the setup and data interpretation. Since CCE is an approximation, it comes with an extra burden of carefully checking its

convergence. Exact numerical modelling is easier to setup computationally and doesn't involve any truncation parameters or other "tuning knobs", allowing for straightforward interpretation. On the other hand, we have access to a very "clean" (strain-engineered) nuclear spin system of a real quantum dot, which yields exact results for a system of $N \sim 10000$ spins. (One can view the real quantum dot as a specialized quantum simulator for studying dynamical decoupling.) From that perspective, implementing CCE would have limited benefit, at least in our particular research problem.

From a broader perspective, this is an interesting question. We anticipate that application of CCE (or similar methods) may encounter challenges that can be formulated as follows. The main philosophy of the approximate expansion methods is to keep the leading orders of interactions while truncating the small residual interactions. The dynamical decoupling works essentially in reverse: it averages out the leading interaction terms to zero so that the resulting spin dynamics are governed by the residual weak interactions. The better the dynamical decoupling sequence, the more complex is the nature of the residual interaction terms and the larger is the number of such residual terms. Thus, the better the dynamical sequence, the more difficult it would be to achieve convergence in an approximate computation method. There are signs of that issue already reported in the numerical tests of CCE [PRB 78 085315 (2008)]: under certain conditions, even going from free induction decay to a 2-pulse Carr-Purcell decoupling required increasing the cluster size from 2-spin to 6-spin. With the 40-pulse decoupling sequence, one may expect to have to include correlations of much higher orders to achieve convergence. The problem of efficient theoretical modelling of dynamical coupling certainly deserves attention, but is somewhat outside the scope of the present work. We comment on these points in the revised Supplementary Note 6.

Reviewer #3:

This manuscript proposes and demonstrates a novel approach to constructing a long-lived quantum memory by utilizing a subspace of the nuclear spin ensemble in a quantum dot. Notably, the memory is encoded into a homogeneous subspace by exploiting strain-induced perturbations in the nuclear spins. As a result, intrinsically long coherence times are successfully achieved. Furthermore, a practical dynamical decoupling (DD) protocol optimized for this system is designed, leading to an experimental demonstration of the longest coherence time to date. These results are accompanied by highly precise and extensive theoretical analysis, which adds depth to the understanding of the mechanisms and ensures reproducibility. Overall, this is a high-quality study on both the experimental and theoretical sides and is evaluated as an important advancement toward the practical realization of quantum repeaters. The novel design concept, the supporting data, and the consistency of the theoretical framework all contribute to the impact of this paper.

We thank the Reviewer for the positive comments.

Minor comments:

1. The fidelity of the memory initialization remains unclear. When storing or retrieving a photonic quantum state in the quantum memory, the initialization rate and DD control fidelity are important aspects. I would appreciate clarification, as much as possible, regarding these points.

In this work we deal specifically with dynamical decoupling of the nuclear spin coherent states. The actual initialization and readout of a prototype nuclear spin quantum memory in a GaAs QD (but without dynamical decoupling of the nuclear spins) has been studied recently [Nature Phys 21, 368 (2025)]. A two-way transfer of a coherent state was demonstrated between a single electron spin qubit and its many-body nuclear spin environment with a transfer fidelity of ~68%. In the revised manuscript we have quoted the fidelity achieved recently in [Nature Phys 21, 368 (2025)].

2. The motivation for using CHASE is not clearly described. DD sequences such as XY8 are widely used in general, so I would like the authors to provide some explanation of the advantages of CHASE. Since the nuclear spin ensemble system may have a structural compatibility with CHASE, I believe the authors' perspective on this would deepen the readers' understanding.

The CHASE sequence and sequences such as Carr-Purcell and its derivative XY8 belong to different classes. Pulse sequences consisting of pi pulses, including XY8, are efficient in decoupling interactions that are linear in terms of spin operators (e.g. $\sim I_z$). These interactions include inhomogeneous chemical shifts and quadrupolar interactions, as well as inhomogeneities and fluctuations of the external magnetic fields. However, sequences of pi pulses are inefficient in decoupling of the spin-spin interactions which are quadratic in terms of spin operators (e.g. $\sim I_{z,1}I_{z,2}$ for spins 1 and 2). In many scenarios sequences such as XY8 may offer excellent decoupling performance, for example in silicon or diamond-based devices where nuclear spins are dilute and spin-spin interactions are relatively small. However, simultaneous suppression of both the linear and quadratic interactions is crucial for dynamical decoupling in a dense nuclear spin lattice of GaAs. CHASE belongs to the class of dynamical decoupling sequences that provides such simultaneous decoupling. In the revised manuscript, we stress this distinctive feature of CHASE dynamical decoupling. Furthermore, in order to verify this point, we have performed numerical modelling of dynamical decoupling in a 12-nucleus dipolar-coupled spin ensemble: as expected, XY8 pulse sequence provides no improvement over Hahn echo, and is considerably inferior to CHASE decoupling.